# Socioeconomic position indicators and risk of alcohol-related medical conditions: A national cohort study from Sweden

**Alexis C. Edwards**[1]*, **Sara Larsson Lönn**[2], **Karen G. Chartier**[1,3], **Séverine Lannoy**[1], **Jan Sundquist**[2,4,5], **Kenneth S. Kendler**[1‡], **Kristina Sundquist**[2,4,5‡]

1 Virginia Institute for Psychiatric and Behavioral Genetics, Department of Psychiatry, Virginia Commonwealth University, Richmond, Virginia, United States of America, 2 Center for Primary Health Care Research, Lund University, Malmö, Sweden, 3 School of Social Work, Virginia Commonwealth University, Richmond, Virginia, United States of America, 4 Department of Family Medicine and Community Health, Icahn School of Medicine at Mount Sinai, New York, New York, United States of America, 5 Department of Population Health Science and Policy, Icahn School of Medicine at Mount Sinai, New York, New York, United States of America

‡ These authors share last authorship on this work.
* alexis.edwards@vcuhealth.org

**Data Availability Statement:** Data cannot be shared publicly because they are only available through application to Statistics Sweden (http:// www.scb.se/) for researchers who meet the criteria for access.

## Abstract

### Background

Alcohol consumption contributes to excess morbidity and mortality in part through the development of alcohol-related medical conditions (AMCs, including alcoholic cardiomyopathy, hepatitis, cirrhosis, etc.). The current study aimed to clarify the extent to which risk for these outcomes differs as a function of socioeconomic position (SEP), as discrepancies could lead to exacerbated health disparities.

### Methods and findings

We used longitudinal Swedish national registries to estimate the individual and joint associations between 2 SEP indicators, educational attainment and income level, and risk of AMC based on International Classification of Diseases codes, while controlling for other sociodemographic covariates and psychiatric illness. We conducted Cox proportional hazards models in sex-stratified analyses ($N$ = 1,162,679 females and $N$ = 1,196,659 males), beginning observation at age 40 with follow-up through December 2018, death, or emigration. By the end of follow-up, 4,253 (0.37%) females and 11,183 (0.93%) males had received an AMC registration, corresponding to overall AMC incidence rates among females and males of 2.01 and 5.20, respectively. In sex-stratified models adjusted for birth year, marital status, region of origin, internalizing and externalizing disorder registrations, and alcohol use disorder (AUD) registration, lower educational attainment was associated with higher risk of AMC in both females (hazard ratios [HRs] = 1.40 to 2.46 for low- and mid-level educational attainment across 0 to 15 years of observation) and males (HRs = 1.13 to 1.48). Likewise, risk of AMC was increased for those with lower income levels (females: HRs = 1.10 to 5.86; males: HRs = 1.07 to 6.41). In secondary analyses, we further adjusted for aggregate familial risk of

**Funding:** This project was supported by grant AA023534 from the US National Institutes of Health to KK and KS, and grants from the Swedish Research Council to JS (2020-01175) as well as ALF funding from Region Skåne awarded to KS. The funders had no role in study design, data collection and analysis, decision to publish, or preparation of the manuscript.

**Competing interests:** The authors have declared that no competing interests exist.

**Abbreviations:** AD, anxiety disorder; AMC, alcohol-related medical condition; ATC, Anatomical Therapeutic Chemical; AUD, alcohol use disorder; DUD, drug use disorder; ED, externalizing disorder; HR, hazard ratio; ID, internalizing disorder; MD, major depression; RERI, relative excess risk due to interaction; SEP, socioeconomic position.

AUD by including family genetic risk scores for AUD ($FGRS_{AUD}$), estimated using medical, pharmacy, and criminal registries in extended families, as covariates. While $FGRS_{AUD}$ were associated with risk of AMC in adjusted models (HR = 1.17 for females and HR = 1.21 for males), estimates for education and income level remained largely unchanged. Furthermore, $FGRS_{AUD}$ interacted with income level, but not education level, such that those at higher familial liability to AUD were more susceptible to the adverse effect of low income. Limitations of these analyses include the possibility of false negatives for psychiatric illness registrations, changes in income after age 40 that were not accounted for due to modeling restrictions, restriction to residents of a high-income country, and the inability to account for individual-level alcohol consumption using registry data.

## Conclusions

Using comprehensive national registry data, these analyses demonstrate that individuals with lower levels of education and/or income are at higher risk of developing AMC. These associations persist even when accounting for a range of sociodemographic, psychiatric, and familial risk factors. Differences in risk could contribute to further health disparities, potentially warranting increased screening and prevention efforts in clinical and public health settings.

## Author summary

### Why was this study done?

- Alcohol consumption contributes to worldwide excess morbidity and mortality, including conditions directly related to alcohol (alcohol-related medical conditions, AMCs) such as alcoholic liver disease.

- The extent to which socioeconomic position (SEP) is longitudinally associated with AMC risk is unclear, but important to understand as it may contribute to health disparities.

### What did the researchers do and find?

- We employed a model that follows people over time to estimate their risk of AMC as a function of 2 SEP indicators: educational attainment and income.

- The model adjusted for other factors that might also be related to risk of AMC, such as marital status, history of psychiatric illness, and aggregate genetic liability to alcohol use disorder (AUD).

- We found that lower educational attainment and lower income were both associated with higher risk of AMC, even after accounting for other predictors.

- Individuals with higher genetic liability to AUD were at increased risk for AMC; these individuals were also more vulnerable to the negative effects of low income.

## What do these findings mean?

- These results indicate that individuals with lower levels of educational attainment or lower incomes might warrant particular clinical attention with respect to screening or evaluation of alcohol consumption, as their risk of AMC is elevated. These differences in risk could exacerbate poor health outcomes among people with lower SEP.

- Clinicians should also be aware that individuals with a family history of AUD are at higher risk of alcohol-related medical morbidities.

- Limitations include the inability to account for individual differences in alcohol consumption, potential for excluding less severe AUD cases, and inability to extend findings to less wealthy countries.

## Introduction

Excessive alcohol use and alcohol use disorder (AUD) contribute substantially to morbidity and mortality worldwide. The World Health Organization estimates that harmful alcohol use accounts for 5.1% of the global burden of disease and injury, resulting in 3 million worldwide deaths per year [1]. In the United States, 9.8% of deaths among working-age adults were attributable to excessive alcohol use [2]. In Sweden, the 2016 prevalence of heavy episodic drinking was 28.0% (12.4% in females and 43.7% in males), slightly lower than in the European Union overall; furthermore, alcohol accounted for 17.6% of unintentional and 21.1% of intentional injuries [3]. The social and psychological consequences of problematic use are considerable, with AUD being linked to higher rates of divorce [4], criminality [5], depression [6], and suicidal behavior [7,8]. Furthermore, excessive alcohol use exacts a steep economic toll: In 2010, the cost of excessive drinking in the United States was estimated at $249 billion [9]. Importantly, alcohol-related medical conditions (AMCs) constituted a substantial proportion of the economic burden of alcohol use, with 11.4% of the 2010 costs attributed to healthcare [9].

Socioeconomic indicators are frequently correlated with alcohol outcomes, including frequency of use, AUD, and alcohol-related morbidity and mortality, and these associations exist within the context of a complex ecosystem of influences [10]. While some studies indicate that drinking frequency and quantity are higher among those with higher socioeconomic position (SEP) [10], others have found that high-frequency/low-quantity drinking is common among those with high SEP, whereas individuals of low SEP drink less frequently but at higher quantities [11–14]. Multiple indicators of lower SEP were associated with increased risk of alcohol-related mortality in a meta-analysis [15], and van Oers [11] reported that lower levels of educational attainment were associated with alcohol-related problems in the Netherlands. In the Swedish Public Health Cohort, lower occupational class was associated with higher risk of alcohol-related health problems, and the effect of heavy episodic drinking was more pronounced among those of lower occupational class [16]. Thus, even when controlling for particularly unhealthy drinking patterns, the extant literature generally suggests that individuals with lower SEP are more likely to suffer severe alcohol-related outcomes.

While many studies report cross-sectional associations between SEP indicators and alcohol outcomes, far fewer have examined longitudinal associations. Clarifying longitudinal associations is particularly important given the possibility of bidirectional relationships [10]: Not only might low SEP lead to poor alcohol outcomes, but alcohol problems might lead to negative

economic consequences [17]. Understanding directionality can be informative to prevention and intervention efforts, i.e., if alcohol-related harm-reduction is the goal, it is necessary to confirm that low SEP is prospectively associated with adverse alcohol outcomes to determine whether targeted prevention could be effective within this group.

In the current study, we aimed to clarify the association between SEP and AMC through the use of longitudinal, nationwide Swedish registry data and Cox proportional hazards models. We examined the associations between 2 SEP indicators—namely, educational attainment and income level—and subsequent risk of developing AMC. We sought to substantively contribute to the existing literature on this topic through the use of large, minimally biased, representative data, which also allows for stratification by sex. We account for a range of potentially important socio-demographic and psychological covariates and confounders. Furthermore, we evaluate whether associations between SEP indicators and AMC shift across adulthood. We hypothesized that lower SEP indicators would be associated with higher risk of AMC and that these associations would be attenuated but persist after accounting for covariates and potential confounders.

## Materials and methods

### Ethics statement

All analyses were approved by the Regional Ethical Review Board of Lund University (no. 2008/409 and later amendments).

### Sample

We used several Swedish nationwide registers, linked using the unique 10-digit identification number assigned at birth or immigration to all Swedish residents. The identification number was replaced by a serial number to ensure anonymity. The registry resources are described in S1 Methods, and included the population register, multi-generation register, multiple health care registers, and the Longitudinal Integrated Database for Health Insurance and Labour Market Statistics (LISA), which included information on income and education. To maximize registry coverage based on preliminary analyses, we included females and males, born between 1950 and 1970, and residing in Sweden at age 40, without a prior AMC registration. Due to missing data on education, $N = 916$ females and $N = 845$ males from the cohort were excluded; due to missing data on marital status, $N = 15$ females and $N = 11$ males were excluded.

### Measures

Our outcome variable, AMC, was defined from Swedish medical registers by the following ICD codes, described further in S1 Methods: ICD8: 571.0, ICD9: 357F, 425F, 535D, 571A, 571B, 571C, 571D; and ICD10: E24.4, G31.2, G62.1, G72.1, I42.6, K29.2, K700, K701, K702, K703, K704, K709, K85.2, K86.0, and O35.4. Individuals with AMC registrations prior to age 40 were excluded.

The primary independent variables of interest were education and familial income. Income was assessed at age 40 and categorized based on the income quartiles for the working-age population in Sweden (ages 20 to 65), although analyses included both employed and unemployed individuals. Educational attainment was categorized into low (compulsory school only), mid (upper secondary school), and high (university level), based on the subject's highest level of school completion. These variables allow us to jointly account for, and distinguish the impact of, indices of individual-level SEP (educational level) and access to resources at the family level (familial income). An individual could have low educational attainment, and/or no personal income, but still have access to resources through a spouse's income.

Sociodemographic covariates included year of birth, marital status, and region of origin. Marital status was assessed at age 40 and obtained from the total population register. The categories were: unmarried, married, divorced, or widowed. Region of origin was defined by country of birth and categorized into: Finland, Western Countries, Eastern Europe, Latin America and the Caribbean, Middle East and Northern Africa, Africa (excluding Northern Africa), and Asia (excluding Middle East) and Oceania. Finland was examined separately because they represent one of the largest immigrant groups in Sweden and because they have higher AUD rates.

From Swedish medical registers, AUD was defined by the following ICD codes: ICD8: 291, 303; ICD9: 305A, 291, 303; and ICD 10: ICD-10 codes: F10.1, F10.2, F10.3, F10.4, F10.5, F10.6, F10.7, F10.8, and F10.9. Additionally, AUD registrations were identified using the Prescribed Drug Register by retrieving "Drugs used in alcohol dependence" (Anatomical Therapeutic Chemical [ATC] Classification System code N07BB); this includes disulfiram (N07BB01), acamprosate (N07BB03), naltrexone (N07BB04), or nalmefene (N07BB05). Finally, we identified AUD cases as those convicted for or suspected of at least 2 alcohol-related crimes according to law 1951:649, paragraph 4 and 4A and law 1994:1009, Chapter 20, paragraphs 4 and 5 from the Swedish Crime Register, and code 3005 and 3201 in the Suspicion register. Most of these crime types include driving or operating a boat while intoxicated. Among criminal AUD registrations, 99.7% were related to drunk driving; however, only 18.3% of AUD registrations are detected using only in the criminal registries.

Internalizing disorder (ID) was defined using the ICD codes for major depression (MD); ICD-9: 296B, 298A, 300E; and ICD-10: F32 and F33, and anxiety disorder (AD); ICD-10: F41. Externalizing disorder (ED) was defined as criminal behavior including white collar crime, property crime, or violent crime as described in detail elsewhere [18]; or drug use disorder (DUD). DUD was identified from medical registers using codes from ICD-9: 292, 304, and 305C - 305I; ICD-10: F11-F16 and from the Prescribed Drug Register if a person had retrieved (on average) more than 4 defined daily doses a day for 12 months of a drug to treat drug dependence (excluding those suffering from cancer), defined by the following ATC codes: N02A, N05C, N05BA, or N07BC. We further defined DUD from the Suspicion Register by codes 3070, 5010, 5011, and 5012; and the Crime Register if convicted by law 1968:64, paragraph 1, point 6 or and law 1951:649, paragraph 4, subsection 2 and paragraph 4A, subsection 2.

Finally, aggregate genetic liability for AUD, which was used in a series of secondary analyses, was operationalized using the family genetic risk score for AUD (FGRS$_{AUD}$), which assesses genetic risk by an examination of first through fifth degree relatives, correcting for cohabitation effects which could make relatives more similar in their AUD risk. This measure is a relative measure which is operationalized as a continuous, standardized indicator of aggregate genetic risk, so that 1 unit corresponds to 1 standard deviation in the population. The details of this measure have been described in detail elsewhere [19]; FGRS have been validated in subsequent research [20].

## Statistical analyses

These analyses were not prospectively described in an analysis plan. S1 Methods provides information on the timing of analyses. Cox proportional hazard models were utilized to estimate the time to AMC, censoring at end of follow-up (December 2018), death, or emigration. Models were stratified by sex to facilitate direct comparisons and enhance data transparency [21].

We conducted a series of preliminary analyses to decide whether time-varying coefficients were necessary for the socioeconomic predictors. This was investigated by including a linear

interaction between follow-up time and education or income. If the interaction was significant and the modification of the magnitude of the hazard ratio (HR) was potentially meaningful, we elected to allow for a time-dependent coefficient in the final models.

We then pursued model building to estimate the association between education and/or income with AMC and investigated whether observed associations were robust to the inclusion of ID, ED, and AUD, which were included as time-varying variables. These models included birth year, marital status, and region of birth as covariates. In Models 1A and 1B, we tested the association between education or income, respectively, with AMC, otherwise including only the sociodemographic covariates. In Model 2, we jointly estimated these associations, including both education and income. In Model 3, we added registrations for ID and ED, as these could act as important mediators between education/income and AMC. Finally, in Model 4, to provide insight into whether education/income is associated with risk for AMC above and beyond variance that could be accounted for by the association between socioeconomic indicators and AUD, we also included AUD as a time-varying variable.

In a secondary set of analyses, we included genetic risk of AUD, operationalized as $FGRS_{AUD}$, to understand how genetic liability to AUD might influence the associations with the investigated risk factors. $FGRS_{AUD}$ is estimated with greater precision for Swedish born subjects with 2 Swedish-born parents, as these individuals have more family members whose data is available in the Swedish registries. Therefore, this series of analyses included only Swedish-born individuals with 2 Swedish-born parents. We ran the same models as above, excluding region of birth. These models are referred to as Model S1A, S1B, S2, S3, and S4, respectively.

We ran an additional model, Model S5, including an interaction between $FGRS_{AUD}$ and education and between $FGRS_{AUD}$ and income, to test whether genetic liability to AUD moderated the associations between education/income and AMC. To be able to interpret the results of the interactions from these multiplicative models on an additive scale, which is often of greater relevance in a public health context [22], we assessed additivity by estimating the relative excess risk due to interaction (RERI) and synergy index (S) [23].

This study is reported as per the Strengthening the Reporting of Observational Studies in Epidemiology (STROBE) guidelines (S1 STROBE Checklist).

## Results

### Descriptive statistics

Tables 1 and 2 provide descriptions of the female and male cohorts, respectively, in the sample of N = 2,359,338 individuals. In the female cohort, N = 4,253 individuals had an AMC registration, corresponding to 0.37% of the sample. Among males, N = 11,183 (0.93%) had an AMC registration. For both sexes, the distribution of sociodemographic variables, including the primary predictors of interest, varied significantly across the groups without and with AMC. For example, individuals with AMC were less likely to be married than their counterparts without AMC (females: 42.04% versus 56.42%; males: 32.59% versus 50.75%). In addition, among individuals residing in Sweden who were born elsewhere, AMC registrations were disproportionately common only among those from Finland: 2.81% of females and 2.06% of males in the cohort were Finnish, but Finns represented 6.51% of females and 5.54% of males with AMC. In contrast, immigrants from Asia, Africa, and the Middle East had disproportionately few AMC registrations. Incidences of AMC for each variable, per 10,000 person years, are provided in the S1 Table.

The polychoric correlation between income quartile and education level for females was 0.185 (SE-0.001) and for males was 0.198 (SE = 0.001), suggesting that while these

**Table 1. Descriptive information on the female cohort, born 1950–1970, with observation beginning at age 40.**

| | Total female cohort (within column %/SD)[1] | Without AMC (within column %/SD)[1] | With AMC (within column %/SD)[1] | With AMC (within row %/SD)[2] | Difference test[3] p-value across those without versus with AMC |
|---|---|---|---|---|---|
| All | 1,162,679 | 1,158,426 | 4,253 | 4,253 (0.37%) | n/a |
| Unmarried | 348,653 (29.99%) | 347,278 (29.98%) | 1,375 (32.33%) | 1,375 (0.39%) | <0.00001 |
| Married | 655,346 (56.37%) | 653,558 (56.42%) | 1,788 (42.04%) | 1788 (0.27%) | |
| Divorced | 152,257 (13.1%) | 151,208 (13.05%) | 1,049 (24.66%) | 1,049 (0.69%) | |
| Widowed | 6,423 (0.55%) | 6,382 (0.55%) | 41 (0.96%) | 41 (0.64%) | |
| Low education | 126,378 (10.87%) | 125,395 (10.82%) | 983 (23.11%) | 983 (0.78%) | <0.00001 |
| Mid education | 551,231 (47.41%) | 549,021 (47.39%) | 2,210 (51.96%) | 2,210 (0.4%) | |
| High education | 485,070 (41.72%) | 484,010 (41.78%) | 1,060 (24.92%) | 1,060 (0.22%) | |
| Low income | 150,952 (12.98%) | 149,596 (12.91%) | 1,356 (31.88%) | 1,356 (0.9%) | <0.00001 |
| Low-mid income | 275,066 (23.66%) | 273,897 (23.64%) | 1,169 (27.49%) | 1,169 (0.42%) | |
| High-mid income | 352,095 (30.28%) | 351,197 (30.32%) | 898 (21.11%) | 898 (0.26%) | |
| High income | 384,566 (33.08%) | 383,736 (33.13%) | 830 (19.52%) | 830 (0.22%) | |
| Africa | 6,372 (0.55%) | 6,363 (0.55%) | 9 (0.21%) | 9 (0.14%) | <0.00001 |
| Asia | 21,140 (1.82%) | 21,119 (1.82%) | 21 (0.49%) | 21 (0.1%) | |
| Eastern Europe | 43,841 (3.77%) | 43,707 (3.77%) | 134 (3.15%) | 134 (0.31%) | |
| Finland | 32,683 (2.81%) | 32,406 (2.8%) | 277 (6.51%) | 277 (0.85%) | |
| Latin America/ Caribbean | 9,299 (0.8%) | 9,279 (0.8%) | 20 (0.47%) | 20 (0.22%) | |
| Middle East/North Africa | 24,546 (2.11%) | 24,531 (2.12%) | 15 (0.35%) | 15 (0.06%) | |
| Sweden | 1,003,382 (86.3%) | 999,681 (86.3%) | 3,701 (87.02%) | 3,701 (0.37%) | |
| Western countries | 21,416 (1.84%) | 21,340 (1.84%) | 76 (1.79%) | 76 (0.35%) | |
| Internalizing registrations | 286,160 (24.61%) | 284,452 (24.56%) | 1,708 (40.16%) | 1,708 (0.6%) | <0.00001 |
| Externalizing registrations | 101,851 (8.76%) | 100,384 (8.67%) | 1,467 (34.49%) | 1,467 (1.44%) | <0.00001 |
| AUD registrations | 40,717 (3.5%) | 38,169 (3.29%) | 2,548 (59.91%) | 2,548 (6.26%) | <0.00001 |
| Mean follow-up time (years) | 18.36 (6.10) | 18.36 (6.10) | 13.61 (6.51) | | <0.00001 |
| Range follow-up time (years) | 0.1, 29.0 | 0.1, 29.0 | 0.1, 28.9 | | |
| Mean birth year | 1960.12 (6.04) | 1960.13 (6.04) | 1957.09 (5.32) | | <0.00001 |
| Range birth year | 1950, 1970 | 1950, 1970 | 1950, 1970 | | |

[1] For reported percentages, the numerator is the N listed in the cell, and the denominator is the N in the top row of the same column. For example, there were N = 348,653 unmarried females, which corresponds to 29.99% of the sample (348,653/1,162,679).

[2] For reported percentages, the numerator is the N listed in the cell, and the denominator is the N in the first column of the same row. For example, there were N = 1,375 unmarried females with AMC, which corresponds to 0.39% of the unmarried females in the sample (1,375/348,653).

[3] For categorical variables, this is a Chi-square; for continuous variables, this is a t test.

AMC, alcohol-related medical conditions; AUD, alcohol use disorder; SD, standard deviation.

socioeconomic measures are related, they also provide nonoverlapping information. The S2 and S3 Tables provide cross-tabulations for females and males, respectively, of income quartile and education level.

## Crude models

In crude models unadjusted for any covariates, females with lower educational attainment were at higher risk of AMC (low versus high: HR = 4.48, 95% CIs = 3.66, 5.47, Chi-square

**Table 2. Descriptive information on the male cohort, born 1950–1970, with observation beginning at age 40.**

| | Total male cohort (col %/SD)[1] | Without AMC (col %/SD)[1] | With AMC (col %/SD)[1] | With AMC (row %/SD)[2] | Difference test[3] |
|---|---|---|---|---|---|
| All | 1,196,659 | 1,185,476 | 11,183 | 11,184 (0.93%) | n/a |
| Unmarried | 475,629 (39.75%) | 470,065 (39.65%) | 5,564 (49.75%) | 5,564 (1.17%) | <0.00001 |
| Married | 605,286 (50.58%) | 601,641 (50.75%) | 3,645 (32.59%) | 3,645 (0.6%) | |
| Divorced | 113,865 (9.52%) | 111,920 (9.44%) | 1,945 (17.39%) | 1,945 (1.71%) | |
| Widowed | 1,879 (0.16%) | 1,850 (0.16%) | 29 (0.26%) | 29 (1.54%) | |
| Low education | 188,732 (15.77%) | 185,959 (15.69%) | 2,773 (24.80%) | 2,773 (1.47%) | <0.00001 |
| Mid education | 580,381 (48.50%) | 574,971 (48.50%) | 5,410 (48.38%) | 5,410 (0.93%) | |
| High education | 427,546 (35.73%) | 424,546 (35.81%) | 3,000 (26.83%) | 3,000 (0.7%) | |
| Low income | 199,405 (16.66%) | 194,905 (16.44%) | 4,500 (40.24%) | 4,500 (2.26%) | <0.00001 |
| Low-mid income | 284,353 (23.76%) | 281,694 (23.76%) | 2,659 (23.78%) | 2,659 (0.94%) | |
| High-mid income | 375,414 (31.37%) | 373,129 (31.47%) | 2,285 (20.43%) | 2,285 (0.61%) | |
| High income | 337,487 (28.2%) | 335,748 (28.32%) | 1,739 (15.55%) | 1,739 (0.52%) | |
| Africa | 7,735 (0.65%) | 7,698 (0.65%) | 37 (0.33%) | 37 (0.48%) | <0.00001 |
| Asia | 15,810 (1.32%) | 15,739 (1.33%) | 71 (0.63%) | 71 (0.45%) | |
| Eastern Europe | 33,009 (2.76%) | 32,759 (2.76%) | 250 (2.24%) | 250 (0.76%) | |
| Finland | 24,624 (2.06%) | 24,004 (2.02%) | 620 (5.54%) | 620 (2.52%) | |
| Latin America/Caribbean | 8,718 (0.73%) | 8,664 (0.73%) | 54 (0.48%) | 54 (0.62%) | |
| Middle East/North Africa | 35,680 (2.98%) | 35,552 (3%) | 128 (1.14%) | 128 (0.36%) | |
| Sweden | 1,047,228 (87.51%) | 1,037,379 (87.51%) | 9,849 (88.07%) | 9,849 (0.94%) | |
| Western countries | 23,855 (1.99%) | 23,681 (2%) | 174 (1.56%) | 174 (0.73%) | |
| Internalizing registrations | 177,998 (14.87%) | 174,804 (14.75%) | 3,194 (28.57%) | 3,194 (1.79%) | <0.00001 |
| Externalizing registrations | 276,156 (23.08%) | 269,886 (22.77%) | 6,270 (56.07%) | 6,270 (2.27%) | <0.00001 |
| AUD registrations | 109,221 (9.13%) | 101,499 (8.56%) | 7,722 (69.05%) | 7,722 (7.07%) | <0.00001 |
| Mean follow-up time (years) | 18.17 (6.03) | 18.01 (6.11) | 13.47 (6.55) | | <0.00001 |
| Range follow-up time (years) | 0.1, 29.0 | 0.1, 29.0 | 0.1, 28.9 | | |
| Mean birth year | 1960.17 (6.03) | 1860.20 (6.03) | 1957.02 (5.26) | | <0.00001 |
| Range birth year | 1950, 1970 | 1950, 1970 | 1950, 1970 | | |

[1]For reported percentages, the numerator is the *N* listed in the cell, and the denominator is the *N* in the top row of the same column. For example, there were
*N* = 475,632 unmarried males, which corresponds to 39.75% of the sample (475,632/1,196,663).

[2]For reported percentages, the numerator is the *N* listed in the cell, and the denominator is the *N* in the first column of the same row. For example, there were *N* = 5,564
unmarried males with AMC, which corresponds to 1.17% of the unmarried males in the sample (5,564/475,632).

[3]For categorical variables, this is a Chi-square; for continuous variables, this is a *t* test.

AMC, alcohol-related medical conditions; AUD, alcohol use disorder; SD, standard deviation.

$p < 0.001$; mid versus high: HR = 1.98, 95% CIs = 1.66, 2.34, $p < 0.001$); the same was true for males, at lower effect sizes (low versus high: HR = 2.27, 95% CIs = 2.01, 2.55, $p < 0.001$; mid versus high: HR = 1.46, 95% CIs = 1.32, 2.61, $p < 0.001$). Likewise, females in lower income quartiles had higher risk of AMC (income quartile 1 versus 4: HR = 11.52 [9.30, 14.27], $p < 0.001$; income quartile 2 versus 4: HR = 3.90 [3.12, 4.87], $p < 0.001$; income quartile 3 versus 4: HR = 1.60 [1.26, 2.03], $p = 0.001$). Results for males were comparable (income quartile 1 versus 4: HR = 11.95 [10.41, 13.72], $p < 0.001$; income quartile 2 versus 4: HR = 3.32 [2.86, 3.86], $p < 0.001$; income quartile 3 versus 4: HR = 1.63 [1.39, 1.91], $p < 0.001$).

## Multivariable models

We next estimated the association between educational attainment (in Model 1A) or income at age 40 (in Model 1B) with risk of AMC, adjusting only for sociodemographic covariates. We

Time dependent HR

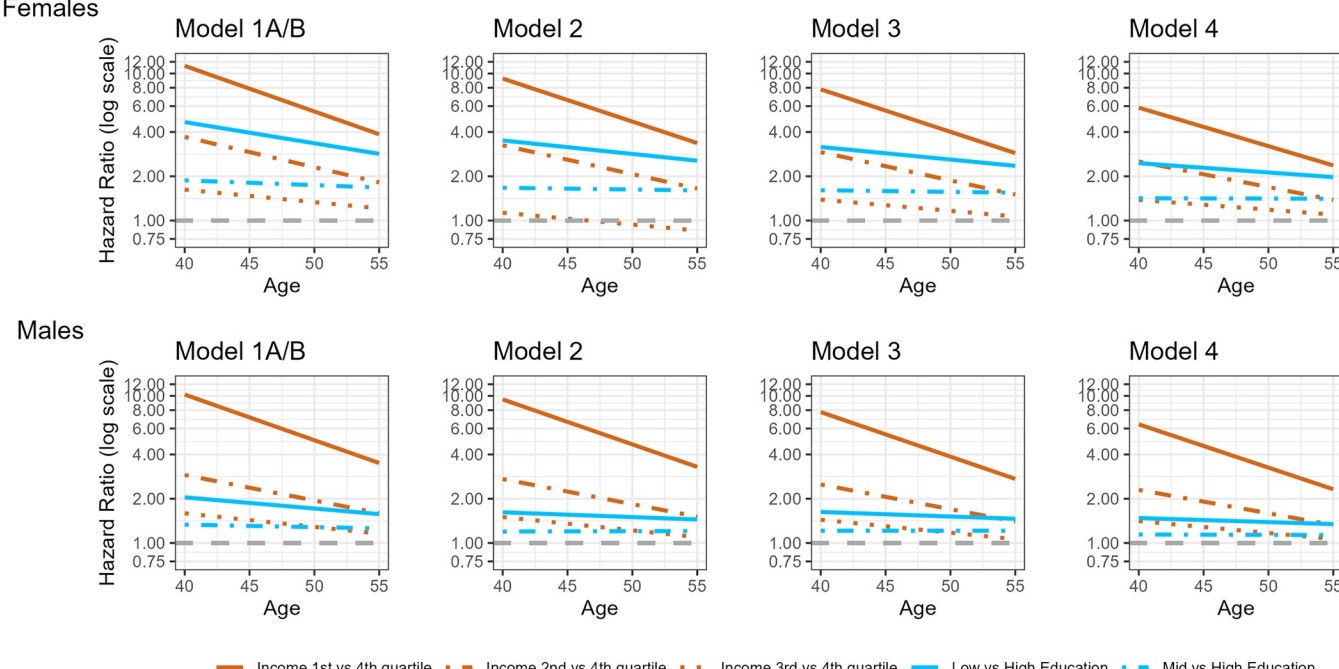

**Fig 1. HRs between education level and/or income with risk of AMCs, across time, from Models 1A/1B–Model 4.** Models 1A and 1B include education level or income, respectively, as the primary predictors of interest, and are adjusted for birth year, marital status, and region of origin. Model 2 includes both education and income level with the aforementioned covariates. Model 3 further adjusts for internalizing and externalizing registrations. Model 4 further adjusts for AUD registrations. Results for females are presented in the top panel and for males in the bottom panel. The horizontal dashed gray line at an HR of 1 represents the null hypothesis. AMC, alcohol-related medical condition; AUD, alcohol use disorder; HR, hazard ratio.

provide snapshots of the HRs for education and income at 4 time points: time 0 (age 40), after 5 years of observation (age 45), after 10 years of observation (age 50), and after 15 years (age 55 or older). As shown in the **Fig 1,** in which estimates from Models 1A and 1B are superimposed for ease of illustration, individuals with lower levels of education or income at age 40 were at substantially increased risk for AMC. Complete results, including for all covariates, are provided in S4 and S5 Tables.

Among females (Fig 1 and S4 Table), having the lowest level of education was associated with a nearly 5-fold increase in risk of AMC in Model 1A (HR = 4.67; 95% CI 3.82, 5.72; $p < 0.001$). Relative to individuals with the highest level of education, those with mid- or low-level education at age 40 were still at increased risk of AMC 15 years later (HRs = 1.68 [95% CIs 1.56, 1.81], $p < 0.001$; and 2.84 [95% CIs 2.60, 3.11], $p < 0.001$, respectively). HRs for the 3 lower quartiles of income were also elevated (Model 1B; Fig 1 and S5 Table). The magnitude of effect declined across observation time, most notably for the lowest quartile (from HR = 11.26 at observation onset to HR = 3.86, 15 years later), and was more pronounced for income than for education. We observed a general dose-response pattern for levels of education/income and for time elapsed since the start of observation; i.e., the strongest effects were observed at the start of observation, and at the lowest education and income levels.

HRs were lower, but still consistently >1, for males (**Fig 1** and S4 Table). In males, being in the lowest education category (Model 1A) was associated with a 2-fold increase in risk of AMC (HR = 2.04, 95% CIs 1.81, 2.30; $p < 0.001$) at the start of observation. Over time, these

estimates declined, but remained above 1 (HRs = 1.26 to 1.57 for those with mid-level and low-level education). Similar to females, the effect size for being in the lowest income quartile was high: HR = 10.21 (95% CIs 8.86, 11.77; $p < 0.001$; Model 1B; Fig 1 and S5 Table). Individuals in the 2 middle quartiles were at lower, but still elevated, risk of AMC. Also consistent with females, the HRs for education were more stable over time than those for income.

In Model 2 (S6 Table), education and income were jointly included as predictors. The HRs were attenuated, but only slightly, indicating that these measures of socioeconomic position account for independent components of AMC risk. We next adjusted for psychiatric illness using time-varying covariates, in Model 3 (S7 Table). Among females, HRs for ID and ED were associated with an approximately 2.6- to 3.7-fold increase in AMC risk. We observed a slight attenuation in the effect sizes of education and income from Model 2. The lowest income quartile remained at high risk of AMC at the start of observation (HR = 7.80, 95% CIs 6.24, 9.75; $p < 0.001$), declining to HR = 2.88, 15 years later. A similar pattern was observed for males, among whom HRs were approximately 2.1 to 3.1 for ID and ED. For males, having a mid- or low-level education was associated with a 1.21- to 1.63-fold increase in AMC risk. Being in the lowest income quartile was still strongly associated with AMC (HR = 7.76, 95% CIs 6.73, 8.95; $p < 0.001$ at start of observation), though the effects were attenuated as time progressed and being in the third quartile 15 years after the onset of observation was not significantly associated with higher AMC risk.

Finally, in Model 4, we adjusted for AUD as a time-varying covariate. AUD was prominently associated with AMC risk: the associated HRs were HR = 34.00 (95% CIs 31.73, 36.42; $p < 0.001$) for females and HR = 10.71 (95% CIs 10.28, 11.15; $p < 0.001$) for males (S8 Table). For both sexes, the inclusion of AUD led to lower HRs for education and income relative to previous models. However, with the exception of the estimate for individuals in the third income quartile, 15 years after the start of observation, all estimates remained significantly >1. We continued to observe a dose-response pattern for income quartiles and as a function of time elapsed since onset of observation in most cases, the exception being that the effect of having mid-level education at age 40 was relatively stable across the observation period. This pattern was clearer among males.

## Secondary analysis

We pursued a secondary set of analyses that corresponded to the original models, replacing region of origin with FGRS$_{AUD}$, as described in the Methods. In Models S1A/S1B through Model S4, the effect of FGRS$_{AUD}$ ranged from HR = 1.17 to 1.39 for females and HR = 1.21 to 1.36 for males (S9–S13 Tables) per standard deviation increase. The HRs for education level and income did not differ substantially from the corresponding HRs from the original Models 1 to 4. We added a model with an interaction term between education/income and FGRS$_{AUD}$, Model S5 to evaluate the possibility that level of aggregate genetic liability to AUD leads to differential susceptibility to the risky effects of lower education or income. The main effects of all predictors and covariates are reported in S14 Table, and we report the RERI and S (estimates of interactions on the additive scale) in S15 Table. For females, there was no evidence that FGRS$_{AUD}$ moderated the association between education level and AMC risk. However, RERI and S estimates were consistent with an interaction between FGRS$_{AUD}$ and the first and second income quartiles, such that individuals at higher genetic liability to AUD were more susceptible to the adverse impact of lower income on risk for AMC. Similarly, in males, we observed no interaction between FGRS$_{AUD}$ and education level, but the association between income and AMC was significantly moderated by FGRS$_{AUD}$.

## Discussion

In the current study, we used nationwide longitudinal Swedish registry data to evaluate a research question of clinical and social importance: whether 2 socioeconomic indicators, education and income, are related to risk of alcohol-related medical conditions. Even after accounting for other sociodemographic measures, psychopathology, and the role of AUD itself, individuals with lower levels of education and income had an increased risk of AMC. For education, the associations were slightly higher among females. For both sexes, the magnitude of effect was higher for lower income level than for lower education, and generally decreased across the observation period but in nearly every case remained elevated even 15 years after the start of observation. Furthermore, the adverse impact of low income was exacerbated among females and males with higher genetic liability to AUD. These findings demonstrate that lower education and income, though related, have at least partially independent and persistent associations with risk of AMC, which could contribute to exacerbated health disparities related to SEP. This effect is only slightly mediated by AUD, suggesting that individuals of lower SEP may warrant particular clinical attention, even within the already high-risk group of individuals with AUD.

The magnitude of effect for income was considerably higher at the onset of observation (age 40) than the corresponding effect of education, and its attenuation was more pronounced across the observation period. The change over time is likely due at least in part to shifts in income during middle to later adulthood: For example, if income increases after age 40, the effect of one's prior income on subsequent health may be superseded by that of their more recent income. However, income at the point of AMC registration might not be the most relevant measure, as AMC is typically the result of chronically high alcohol consumption that began years before.

As noted above, previous research has found that drinking patterns vary by SEP, with those of lower SEP more likely to engage in heavy consumption [11–14,16], which is strongly linked to risk of AMC [24–26]. Information on consumption levels is not available in the Swedish nationwide registries, precluding our ability to test whether the associations between education/income remain when accounting for chronic heavy drinking, for example. However, our finding that these associations persist when controlling for AUD—itself closely associated with consumption levels [27,28]—is consistent with prior evidence that individuals of lower SEP are at disproportionately high risk of AMC even after accounting for heavy drinking [16].

With respect to the discrepancy between effect sizes for education and income, some prior research suggests that education is a more potent risk factor for the onset of chronic health conditions, while income more strongly predicts their progression [29]. A study that included the current cohort reported more protective effects between education and AUD than between income and AUD [30], though differences were not as pronounced as observed in the current study. Although these nuances remain to be investigated in future research, the current study clearly indicates that education and income are independently associated with AMC risk, though in assessing future risk, clinicians should pay particular attention to income.

What might contribute to this unequal risk across socioeconomic strata? Some studies indicate that individuals of lower SEP are less likely to engage in preventive health care behaviors [31]. Even in countries with socialized medical care, where access should be less of a barrier, these discrepancies may be due to health literacy [32] and less expendable time [33]. A study of large Finnish and British cohorts found that the association between lower SEP and poor health outcomes was driven in part by psychiatric and substance use disorders, which in turn led to downstream medical problems [34]. Still, accounting for other AUD registrations in the current study resulted in only slight attenuations to the observed associations, providing evidence that SEP indicators do not merely exert their effects through AUD.

In addition to the effects of our primary predictors of interest, we note 3 incidental findings that merit further investigation in the context of models specifically designed to replicate our observed associations. First, in fully adjusted models (Model 4 and Model S4), unmarried females were at reduced risk of AMC relative to married females, while among males, married individuals were consistently at lowest AMC risk. Prior studies have supported a protective effect of marriage for alcohol use, AUD, and other psychiatric outcomes [35–39], suggesting that AMC is a relative outlier, and only for females. This could be due to a stronger negative impact of husbands' alcohol use on wives' alcohol-related outcomes [40], an exposure that would not apply to unmarried females. The dynamics at play are likely to be challenging to dissect using registry data, but merit follow-up.

Second, with the exception of those from Finland, immigrants were at lower risk of AMC than their Swedish-born counterparts. One potential explanation is that immigrants from many regions have lower levels of alcohol consumption than Swedes and many other European nations [41]. A study of immigrant cohorts in Sweden showed that foreign-born Swedes from Asia and the Middle East (males and females) and from African and Eastern Europe (females) had lower rates of AUD than the general population [42]. Finns, who constitute one of the largest immigrant groups in Sweden, had the highest AUD rates (males and females). Finland is included in the so-called "vodka belt," i.e., those Northern countries where the use of stronger alcoholic beverages (spirits) and binge drinking are more common than in other countries [43]. Prior studies also document Finns' excess mortality in Sweden across multiple medical conditions [44] and their self-reported poor health compared to other Swedes and to Finns living in Finland [45], which together suggests their greater risk of AMC signals wider health inequities. Additionally, immigrant groups may be less likely to use medical care, even in nations where it is broadly accessible. For example, foreign-born patients are less likely than Swedish-born patients to pick up their medication for treating AUD from the pharmacy [46]. It then is possible that registry data could potentially underrepresent the extent of AMC among immigrants. If this is the case, additional outreach may be necessary to ensure that immigrants are adequately screened for risky drinking and treated for AUD to prevent the development of AUD and AMC.

Third, we found that individuals with higher genetic liability to AUD were more susceptible to the adverse impact of low income, though this was not the case for those with lower levels of educational attainment. This could reflect the stronger overall effect size of income level, and potentially due to the fact that the estimate is based on year 0, which is when the effect of income was most pronounced; replication in other samples, and perhaps using complementary methods (e.g., molecular polygenic scores) is needed. However, these results do suggest that clinicians should be attentive to the potentially combinatorial effects of low-income and dense family history of AUD: Patients subject to both conditions may warrant additional screening to ensure that their alcohol use does not result in adverse health outcomes.

Strengths of the current analysis include the use of nationwide, longitudinal databases with minimal bias; comparison of 2 related measures of SEP with unique contributions to AMC risk; and our ability to control for important covariates such as comorbid psychopathology and genetic liability to AUD. Our findings must still be viewed in the context of several limitations. First, income can change frequently, but to facilitate modeling we focused on income at the beginning of observation (age 40). Including repeated measures of income could have resulted in different findings, e.g., the effect size of income might decline less rapidly than in the present study. Second, while individuals with an AMC registration are likely to meet criteria for AUD, approximately 30% to 40% of those with AMC registrations did not have another registration for AUD (e.g., through the ICD-10 F10 codes), suggesting inconsistency in how clinicians record patients' AUD. Third, due to the nature of $FGRS_{AUD}$, these scores are less

precise among immigrants, precluding our ability to confidently estimate the extent to which aggregate genetic liability to AUD contributes directly to AMC and/or mediates/moderates the effects of income or education among those born outside of Sweden. Fourth, we were unable to directly account for differences in alcohol consumption, which is correlated with AUD, and which may differ by SEP. The medical community's perspective on the impact of alcohol consumption at different levels is variable and subject to revision. For example, while many studies support a J-shaped association between consumption and certain health outcomes [47], other considerations (e.g., the climate-based impacts of the alcohol industry) have led some entities to recommend abstinence [48]. These issues are beyond the scope of the current study but warrant further consideration for public health considerations.

Fifth, we cannot firmly ascribe the observed associations to a causal path: Education and income are correlated with an array of psychosocial factors that could act as confounders, and complementary methodological approaches are necessary to provide additional insight. Finally, alcohol consumption varies across countries and cultures, and the current findings might not generalize to other contexts. For example, although the drinking age is as low as 18 in Sweden, alcohol access is constrained by relatively high costs and by the Systembolaget, which controls liquor sales and has limited hours of operation. Greater alcohol accessibility in other countries, especially where costs are lower, could exacerbate the risks of AMC associated with lower SEP. Furthermore, the sex differences observed in the current study might not be replicated in societies with lower sexual egalitarianism. Additional research in other cultural contexts is therefore essential.

In summary, we provide evidence that individuals with lower levels of education and/or income are more likely to suffer from AMC, even after accounting for differences in AUD, comorbid psychopathology, and aggregate genetic liability. Income generally plays a more prominent role, though the effects of both SEP indicators persist across time. Secondary observations suggest that more research is needed to better understand differences related to sex, marital status, and region of origin. These findings contribute to a growing body of literature on health disparities as a function of socioeconomic resources and suggest that individuals with fewer such resources could benefit from additional clinical attention regarding the risks associated with problematic alcohol use.

## Supporting information

**S1 Methods. Description of registry resources and analytic methods.**
(DOCX)

**S1 Table. Incidence rates of alcohol-related medical conditions, reported as number of new cases per 10,000 person years, for variables that are constant over time.**
(DOCX)

**S2 Table.** Cross-tabulation of income quartile and education level for females in the full sample (top panel) and among those with an alcohol-related medical condition (AMC; bottom panel).
(DOCX)

**S3 Table.** Cross-tabulation of income quartile and education level for males in the full sample (top panel) and among those with an alcohol-related medical condition (AMC; bottom panel).
(DOCX)

**S4 Table. Complete results for Model 1A for females and males, testing the association between education level and alcohol-related medical conditions.** Hazard ratios, 95%

confidence intervals, and *p*-values from Chi-square tests are presented. The primary predictor of interest (here, education level) was modeled using a time-varying coefficient, with a linear term for time. Below, we provide snapshots of hazard ratios for education at 4 time points: at the beginning of observation (time 0), after 5 years, after 10 years, and after 15 years. (DOCX)

**S5 Table. Complete results for Model 1B for females and males, testing the association between income and alcohol-related medical conditions.** Hazard ratios, 95% confidence intervals, and *p*-values from Chi-square tests are presented. The primary predictor of interest (here, income) was modeled using a time-varying coefficient, with a linear term for time. Below, we provide snapshots of hazard ratios for income at 4 time points: at the beginning of observation (time 0), after 5 years, after 10 years, and after 15 years. (DOCX)

**S6 Table. Complete results for Model 2 for females and males, testing the associations between education level and income with alcohol-related medical conditions.** Hazard ratios, 95% confidence intervals, and *p*-values from Chi-square tests are presented. The primary predictors of interest (education level and income) were modeled using time-varying coefficients, with a linear term for time. Below, we provide snapshots of hazard ratios for education and income at 4 time points: at the beginning of observation (time 0), after 5 years, after 10 years, and after 15 years. (DOCX)

**S7 Table. Complete results for Model 3 for females and males, testing the associations between education level and income with alcohol-related medical conditions.** Hazard ratios, 95% confidence intervals, and Chi-square *p*-values are presented. The primary predictors of interest (education level and income) were modeled using time-varying coefficients, with a linear term for time. Below, we provide snapshots of hazard ratios for education and income at 4 time points: at the beginning of observation (time 0), after 5 years, after 10 years, and after 15 years. (DOCX)

**S8 Table. Complete results for Model 4 for females and males, testing the associations between education level and income with alcohol-related medical conditions.** Hazard ratios, 95% confidence intervals, and Chi-square *p*-values are presented. The primary predictors of interest (education level and income) were modeled using time-varying coefficients, with a linear term for time. Below, we provide snapshots of hazard ratios for education and income at 4 time points: at the beginning of observation (time 0), after 5 years, after 10 years, and after 15 years. (DOCX)

**S9 Table. Complete results for Model S1A for females and males, testing the association between education level and alcohol-related medical conditions.** Hazard ratios, 95% confidence intervals, and Chi-square *p*-values are presented. The primary predictor of interest (here, education level) was modeled using a time-varying coefficient, with a linear term for time. Below, we provide snapshots of hazard ratios for education at 4 time points: at the beginning of observation (time 0), after 5 years, after 10 years, and after 15 years. These secondary analyses were limited to the subsample born in Sweden with 2 Swedish-born parents to improve the precision of the family genetic risk score for alcohol use disorder; accordingly, region of interest is excluded as a covariate. (DOCX)

**S10 Table. Complete results for Model S1B for females and males, testing the association between income and alcohol-related medical conditions.** Hazard ratios and 95% confidence intervals are presented. The primary predictor of interest (here, income) was modeled using a time-varying coefficient, with a linear term for time. Below, we provide snapshots of hazard ratios for income at 4 time points: at the beginning of observation (time 0), after 5 years, after 10 years, and after 15 years. These secondary analyses were limited to the subsample born in Sweden with 2 Swedish-born parents to improve the precision of the family genetic risk score for alcohol use disorder; accordingly, region of interest is excluded as a covariate. (DOCX)

**S11 Table. Complete results for Model S2 for females and males, testing the associations between education level and income with alcohol-related medical conditions.** Hazard ratios, 95% confidence intervals, and Chi-square *p*-values are presented. The primary predictors of interest (education level and income) were modeled using time-varying coefficients, with a linear term for time. Below, we provide snapshots of hazard ratios for education level and income at 4 time points: at the beginning of observation (time 0), after 5 years, after 10 years, and after 15 years. These secondary analyses were limited to the subsample born in Sweden with 2 Swedish-born parents to improve the precision of the family genetic risk score for alcohol use disorder; accordingly, region of interest is excluded as a covariate. (DOCX)

**S12 Table. Complete results for Model S3 for females and males, testing the associations between education level and income with alcohol-related medical conditions.** Hazard ratios, 95% confidence intervals, and Chi-square *p*-values are presented. The primary predictors of interest (education level and income) were modeled using time-varying coefficients, with a linear term for time. Below, we provide snapshots of hazard ratios for education level and income at 4 time points: at the beginning of observation (time 0), after 5 years, after 10 years, and after 15 years. These secondary analyses were limited to the subsample born in Sweden with 2 Swedish-born parents to improve the precision of the family genetic risk score for alcohol use disorder; accordingly, region of interest is excluded as a covariate. (DOCX)

**S13 Table. Complete results for Model S4 for females and males, testing the associations between education level and income with alcohol-related medical conditions.** Hazard ratios, 95% confidence intervals, and Cho-square *p*-values are presented. The primary predictors of interest (education level and income) were modeled using time-varying coefficients, with a linear term for time. Below, we provide snapshots of hazard ratios for education level and income at 4 time points: at the beginning of observation (time 0), after 5 years, after 10 years, and after 15 years. These secondary analyses were limited to the subsample born in Sweden with 2 Swedish-born parents to improve the precision of the family genetic risk score for alcohol use disorder; accordingly, region of interest is excluded as a covariate. (DOCX)

**S14 Table. Complete results for Model S5 for females and males, testing the associations between education level and income with alcohol-related medical conditions.** Hazard ratios, 95% confidence intervals, and Chi-square *p*-values are presented. The primary predictors of interest (education level and income) were modeled using time-varying coefficients, with a linear term for time. Below, we provide snapshots of hazard ratios for education level and income at 4 time points: at the beginning of observation (time 0), after 5 years, after 10 years, and after 15 years. These secondary analyses were limited to the subsample born in Sweden with 2 Swedish-born parents to improve the precision of the family genetic risk score for

alcohol use disorder; accordingly, region of interest is excluded as a covariate. This model includes an interaction term between $FGRS_{AUD}$ and each level of education and income; those results are in S15 Table.
(DOCX)

**S15 Table. Model S5 includes an interaction term between $FGRS_{AUD}$ and each level of education and income (main effects are provided in S11 Table).** We present these as interactions on the additive scale through the use of the relative excess risk due to interaction (RERI) and synergy index (S), along with corresponding 95% confidence intervals; the *p*-value is based on a Chi-square test of the interaction term. These terms are estimated based on the effect of education and income at time 0.
(DOCX)

**S1 STROBE Checklist. STROBE checklist.**
(DOCX)

## Author Contributions

**Conceptualization:** Alexis C. Edwards, Sara Larsson Lönn.

**Data curation:** Sara Larsson Lönn.

**Formal analysis:** Sara Larsson Lönn.

**Funding acquisition:** Jan Sundquist, Kenneth S. Kendler, Kristina Sundquist.

**Methodology:** Alexis C. Edwards, Sara Larsson Lönn.

**Project administration:** Alexis C. Edwards, Jan Sundquist, Kenneth S. Kendler, Kristina Sundquist.

**Writing – original draft:** Alexis C. Edwards, Sara Larsson Lönn.

**Writing – review & editing:** Alexis C. Edwards, Sara Larsson Lönn, Karen G. Chartier, Séverine Lannoy, Jan Sundquist, Kenneth S. Kendler, Kristina Sundquist.

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
