## [Editor Report · Decision Letter 0]

7 Nov 2023

Dear Dr Edwards, 

Thank you for submitting your manuscript entitled "Lower educational attainment and income level are independently associated with risk of alcohol-related medical conditions: A Swedish national cohort study" for consideration by PLOS Medicine.

Your manuscript has now been evaluated by the PLOS Medicine editorial staff and I am writing to let you know that we would like to send your submission out for external peer review.

Please re-submit your manuscript within two working days, i.e. by Nov 09 2023 11:59PM.

Feel free to email me at pdodd@plos.org (or the team at plosmedicine@plos.org) if you have any queries relating to your submission.

Kind regards,

Pippa

Philippa Dodd, MBBS MRCP PhD

PLOS Medicine

---

## [Decision Letter · Decision Letter 1]

7 Dec 2023

Dear Dr. Edwards,

Thank you very much for submitting your manuscript "Lower educational attainment and income level are independently associated with risk of alcohol-related medical conditions: A Swedish national cohort study" (PMEDICINE-D-23-03228R1) for consideration at PLOS Medicine. 

[LINK]

In light of these reviews, I am pleased to tell you that we would like to consider a revised version that addresses the reviewers' and editors' comments. Obviously we cannot make any decision about publication until we have seen the revised manuscript and your response, and we plan to seek re-review by one or more of the reviewers. 

We expect to receive your revised manuscript by Dec 28 2023 11:59PM. Please email us (plosmedicine@plos.org) if you have any questions or concerns.

We look forward to receiving your revised manuscript. 

Best wishes,

Pippa

Philippa Dodd, MBBS MRCP PhD

PLOS Medicine

plosmedicine.org

pdodd@plos.org

COMMENTS FROM THE ACADEMIC EDITOR

A major revision seems reasonable. I had just two thoughts to add:

1. Maybe I missed this, but why limit the lower end of the age range to 40 years? The listed alcohol-related medical conditions do occur in younger populations. Is it a sample size problem?

2. I wonder if the authors might consider making a conceptual model to visually depict the relationships and confounders they include in the models. I understand that it isn't a fully causal DAG, but I find that such figures tend to help readers a lot. This is not a mandatory comment, just a suggestion for interpretability/readability.

COMMENTS FROM THE EDITORS

GENERAL

Please respond to all editor and reviewer comments detailed below, in full.

Please include line and page numbers to aid reviewing and editing.

Please ensure that the study is reported according to the STROBE guideline, and include the completed STROBE checklist as Supporting Information. Please add the following statement, or similar, to the Methods: "This study is reported as per the Strengthening the Reporting of Observational Studies in Epidemiology (STROBE) guideline (S1 Checklist)."

When completing the checklist, please use section and paragraph numbers, rather than page and/or line numbers as these often change in the event of publication.

COMPETING INTERESTS

All authors must declare their relevant competing interests per the PLOS policy, which can be seen here:

https://journals.plos.org/plosmedicine/s/competing-interests

For authors with ties to industry, please indicate whether any of the interests has a financial stake in the results of the current study.

Please include the competing interest statement only in the manuscript submission form when you re-submit the manuscript an remove from the title page.

ETHICS STATEMENT

Please kindly include as stated, per our previous email correspondence.

TITLE

Please revise your title according to PLOS Medicine's style. Your title must be nondeclarative and not a question. It should begin with main concept if possible. "Effect of" should be used only if causality can be inferred, i.e., for an RCT. Please place the study design ("A randomized controlled trial," "An observational study," "A modelling study," etc.) in the subtitle (ie, after a colon), as in the current version.

We note and thank the reviewer (please see below) for their interest in the genetic risk score and for the comments regarding its reference in title. We agree that it is very interesting, but as it forms part of your secondary analyses, we don’t think it necessary for the title to specifically detail this or for extensive additional details to be added to the abstract.

ABSTRACT

Abstract Background:

Please ensure that the final sentence clearly states the study question.

Abstract Methods and Findings:

Please include (brief) details of the registries used to leverage your data including those used to obtain FGRS.

Please clearly define the length of follow-up (mean, SD, range).

Please provide details of the important dependent variables that you refer to.

Please include the actual amounts and/or absolute risk(s) of relevant outcomes, not only the relative measures (example for absolute risks: PMID: 28399126).

Please quantify the main results with 95% CIs and p values. Please report p as <0.001 and where higher the exact p value as p=0.002, for example. Please use commas as opposed to hyphens to separate upper and lower CI bounds as the latter can be confused with reporting of negative values. 

Please ensure that all abbreviations used for statistical reporting have been defined at first use for the reader.

‘…even after adjusting for all other covariates and potential confounders…’ as above please provide details of the important dependent variables adjusted for.

We don’t think that it is necessary to present both crude and adjusted analyses in the abstract (but it certainly doesn’t hurt to do so) so, if necessary, to conserve space during your revisions you could present only the adjusted results (and the factors adjusted for). 

Please define ‘AUD’ for the reader at first use.

AUTHOR SUMMARY

At this stage, we ask that you include a short, non-technical Author Summary of your research to make findings accessible to a wide audience that includes both scientists and non-scientists. The authors summary should consist of 2-3 succinct bullet points under each of the following headings:

• Why Was This Study Done? Authors should reflect on what was known about the topic before the research was published and why the research was needed.

• What Did the Researchers Do and Find? Authors should briefly describe the study design that was used and the study’s major findings. Do include the headline numbers from the study, such as the sample size and key findings. 

• What Do These Findings Mean? Authors should reflect on the new knowledge generated by the research and the implications for practice, research, policy, or public health. Authors should also consider how the interpretation of the study’s findings may be affected by the study limitations. In the final bullet point of ‘What Do These Findings Mean?’, please describe the main limitations of the study in non-technical language.

The Author Summary should immediately follow the Abstract in your revised manuscript. This text is subject to editorial change and should be distinct from the scientific abstract. Please see our author guidelines for more information: https://journals.plos.org/plosmedicine/s/revising-your-manuscript#loc-author-summary

INTRODUCTION

Final paragraph – suggest revising/moving the sentence beginning ‘Our approach…’ which seems better suited to the discussion as written as written. Reference to stratification by sex should remain in the introduction. 

Penultimate sentence, suggest, ‘We account for a range…’

METHODS and RESULTS

Please also see reviewer comments (below) regarding your methodological approach.

As above, please add the following statement, or similar, to the Methods: "This study is reported as per the Strengthening the Reporting of Observational Studies in Epidemiology (STROBE) guideline (S1 Checklist)."

When completing the checklist, please use section and paragraph numbers, rather than page or line numbers as these often change in the event of publication.

Please also include the ethics statement, as requested above, including the name(s) of the institutional review board(s) that provided ethical approval.

Did your study have a prospective protocol or analysis plan? Please state this (either way) early in the Methods section.

For all observational studies, in the manuscript text, please ensure that you indicate: 

(1) the specific hypotheses you intended to test, 

(2) the analytical methods by which you planned to test them, 

(3) the analyses you actually performed, and 

(4) when reported analyses differ from those that were planned, transparent explanations for differences that affect the reliability of the study's results. If a reported analysis was performed based on an interesting but unanticipated pattern in the data, please be clear that the analysis was data-driven.

Results, opening paragraph – please explicitly state the total number of participants included in the full dataset.

Please clearly define the length of follow up (eg, in mean, SD, and range).

Please ensure that you provide the actual numbers of events for outcomes not just summary statistics or OR/HR.

Please ensure that percentages are quantified with numerators and denominators.

As above for the abstract, please quantify the main results with 95% CIs and p values. Please report p as <0.001 and where higher the exact p value as p=0.002, for example. Please use commas as opposed to hyphens to separate upper and lower CI bounds as the latter can be confused with reporting of negative values. Suggest reporting statistical information as follows, ‘(HR 4.71; 95% CI [3.85, 5.77]; p</=)’. Please check and amend throughout (including supporting information) to ensure consistency of reporting.

Please ensure that all abbreviations used for statistical reporting have been defined at first use for the reader.

When a p value is given, please also include the statistical test used to determine it.

TABLES

Throughout (including the supporting information) please ensure that each table is affiliated to an appropriate caption which clearly explains the table content without the need to refer to the text. Please ensure that any abbreviations (including those used for statistical reporting) are clearly defined within the caption (or a footnote).

Please ensure that all numerical values are clearly defined for the reader (including in the supporting information).

Please report p as <0.001 and where higher the exact p value.

For main outcomes measures where 95% CIs are presented please also present p values, reporting p as <0.001 and where higher the exact p value. 

Throughout (including the supporting information) please ensure that you indicate whether analyses are adjusted or unadjusted. For the purpose of transparent data reporting, where adjusted analyses are presented please present the unadjusted analyses for comparison and clearly detail the factors adjusted in the caption (or footnote)

Tables 1 and 2 – please clearly define ‘col/% SD’ in the column header.

FIGURES

Throughout (including the supporting information), please ensure that each figure is affiliated to an appropriate caption which clearly explains it content without the need to refer to the text. Please ensure that any abbreviations (including those used for statistical reporting) are clearly defined within the caption (or a footnote).

Please see here for guidelines on submitting and citing figures https://journals.plos.org/plosmedicine/s/figures#loc-how-to-submit-figures-and-captions

Please consider avoiding the use of green and/or red to make your figures more accessible to those with color blindness.

DISCUSSION

Please also see reviewer comments

REFERENCES

For in-text reference callouts please place citations in square parentheses separate by commas. For example, [1,3,6] or [1-3]. Please check and amend throughout all sub-sections of the manuscript and supporting files.

In the bibliography please ensure that you list up to but no more than 6 author names followed by et al.

For all web references please ensure you include an, ‘Accessed [date].’

Journal name abbreviations should be those listed in the National Center for Biotechnology Information (NCBI) databases.

SUPPORTING INFORMATION

Throughout please ensure all requirements detailed above for tables, figures, statistical reporting and referencing are applied to the supporting information, as relevant.

Please cite your Supporting Information as outlined here: https://journals.plos.org/plosmedicine/s/supporting-information

In the published article, supporting information files are accessed only through a hyperlink attached to the captions. For this reason, you must list captions at the end of your manuscript file. You may include a caption within the supporting information file itself, as long as that caption is also provided in the manuscript file. Do not submit a separate caption file.

COMMENTS FROM THE REVIEWERS

Reviewer #1: See attachment

Michael Dewey

Reviewer #2: Thank you for the invitation to review this manuscript. I guess it is because I have performed related research previously. 

The study assesses the association of socioeconomic position with alcohol-related medical conditions. The interplay between different measures of alcohol, SEP measures and different health outcomes related to alcohol and mortality has been studied extensively. The authors do not try to oversell the importance of the paper, they are sober, and argue that the study is an important contribution to the literature because of how they made use of different information from registry data. I don't how a complete overview of comparable studies, but I am impressed by how the authors have used the different registries to define alcohol use disorder, alcohol-related medical conditions, and other study variables, as well as the family genetic risk score for AUD. I can see that these registries and the different definitions has been used previously by the authors. 

The manuscript appears to me as being well written and I haven't found anything that I with confidence can say is wrong, although there are some things I would like the authors to clarify. Perhaps the most important is related to the measure of income and the use risk estimates from the final model for predefined covariates, such as to say something about how the risk of AMC differ according to biological sex and background. 

Comment 1

Quoting: "The primary independent variables of interest were education and familial income. Income was assessed at age 40 and categorized based on the income quartiles for the working population, aged 20 to 65, in Sweden."

Are the analyses restricted to individuals who were working or restricted to this population when income is included in the model? Or does the authors mean that income was assessed for individuals in the age range 20 to 65, an age range where people tend to work? I would have liked that the authors commented on why they didn't include "working yes or no" or something related to employment as a covariate.

The authors write in the manuscript that income is "familial income", which I assume is the total income of the household, including both the index person and the partner? Please make this much clearer in the manuscript than what it is in the current. Related to the question above, does this mean that a person could be unemployed, but have an income in the 4th quartile if they have a partner that earns well? Please clarify. 

Consider including some type of discussion regarding the use of familial income versus personal income as a measure of SEP. For example, the authors have chosen personal education level, not total educational level, so there must be a good reason for using household/familial income?

Comment 2:

The authors highlights from the analyses results based on risk estimates of covariates. In other words, they start asking more research questions, such as: Does unmarried women have a higher risk of AMC then married women? Do Swede-Fins have a higher risk of AMC? These research questions or additional analyses should be mentioned earlier in the manuscript, as they are presented as important findings, but were not predefined. 

The authors write that unmarried females were at reduced risk of AMC relative to married females. Now I am not an expert on causal pathways, DAGs, statistics and causality, but I have been told not to take a model specified to answer a causal research question (in this study, what is the association of SEP with AMC), and answer other research questions using the risk estimates from the covariates. If the model is set up for prediction (all variables in, then this is okey I was told). 

I'm not saying it is wrong in this occasion, but the advice is probably to avoid ending up in situations where the model is not well suited to answer research questions pertaining to other variables than the exposure of interest. The other research question here is whether AMC risk differs by marital status, which is answered by a model set up to answer whether the risk of AMC differ by SEP. Please give it a thought, for example, consider whether adjusting the association between marital status and AMC risk for familial income is ok, as this is what is being done now I think. I guess it would place a lot of unmarried women without a partner in the "low income" section, and I don't know if that would be the best SEP indicator. 

Comment 3:

Harmful and problematic alcohol consumption. Alcohol is defined as a Group 1 carcinogen (IARC). So, in that sense, any alcohol intake is harmful in a dose-dependent manner. How it interacts with mental health and emotions is well, difficult, bi-directional and perhaps multidirectional. I mean, alcohol is probably a part of celebrations as well as a bad coping mechanism. 

The "not harmful" version stems from epidemiological studies of the traditional sort, showing that individuals reporting light to moderate, and in some cases also quite frequent intake, has a lower HR of ischemic heart disease in comparison with no/low/infrequent drinking. And because ischemic heart disease caused so many premature deaths among men in particular, and still does, but less so because of primary prevention (smoking and diet) and better treatment (medication and PCI), this association propagated into a lower risk of all-cause death in similar studies. Whether alcohol has a beneficial effect on ischemic heart disease is much more debated and controversial today, mainly because studies using instrumental variables does not find a beneficial association of alcohol with IHD (at least that was the case the last time I checked a couple of years ago), creating a discrepancy between results from "traditional" and "modern" observational studies. In the now published Nordic nutrition recommendations (revised every 8 years), "no alcohol intake" is promoted as the intake level that is associated with the lowest health risk of alcohol, taking into account alcohols effect on cancer, heart, as well as accidents and violence and drunken driving etc. I haven't found the background paper on this, but the conclusion seems to be based on the work with the Canadian low drinking guidelines, where the conclusion is as follows: "the most recent and highest quality systematic reviews showing that drinking a little alcohol neither decreases nor increases the risk of ischemic heart disease, but it is a risk factor for most other types of cardiovascular disease, including, hypertension, heart failure, high blood pressure, atrial fibrillation and hemorrhagic stroke"…

Paradis, C., Butt, P., Shield, K., Poole, N., Wells, S., Naimi, T., Sherk, A., & the Low-Risk Alcohol Drinking Guidelines Scientific Expert Panels. (2023). Canada's Guidance on Alcohol and Health: Final Report. Ottawa, Ont.: Canadian Centre on Substance Use and Addiction

Why am I writing this? The consensus might be moving in the direction that "any alcohol intake is harmful", which would be the case if there really is no beneficial effect of alcohol on the development of ischemic heart disease. Because the manuscript starts with referring to data from WHO regarding harmful alcohol use, and also use the word "problematic" alcohol use, which is in the same category, the authors states that one type of alcohol intake is harmful and indirectly that another type is not harmful. 

I leave it up to the authors to decide whether they want to address this or not, but I encourage them to at least consider it. However, as a reader, I would prefer if the authors were more specific when they refer to literature regarding measures of alcohol intake and measures of health outcomes. When referring to intake as "harmful", what is the definition in the study? When referring to intake as "excessive", what is meant? When referring to a paper on heavy episodic drinking, what is the definition? 

Comment 4:

Introduction, second paragraph: I think I understand what is meant with the term "lower occupational class", but mayhaps "it is better to write "lower socioeconomic position measured by occupation"? 

Comment 5:

Although I understand the need to be short and there could be a word limit, consider whether "alcohol outcomes" is a term that is used extensively as an umbrella term for measures of alcohol intake and measures of health outcomes that is associated with different measures of alcohol intake. 

Comment 6:

Individuals that died before the age of 40 years were not allowed to participate in this study, by design. It is perhaps worth mentioning and it should be commented on whether the authors think it is relevant for the association under study. 

Comment 7:

Figure: Would it be correct to state that you have superimposed the hazard lines for each predictor? 

Figure legend. There is no information in the figure legend that the hazard ratio is the risk of AMC. It now states that the hazard ratio is for education level and income. 

There is no information on what is included in the models, so in order to make use of the figure, the reader must go back to method section. I have always included this information in the legends, but can't find anything about this in strobe guidelines or journal style that says what is correct. You choose I guess. Might be that this part is not included in the pdf I have in front of me. 

Regarding the figures. At glance, I can't see any differences between the 4 figures for women or the 4 for men. If there is a difference, I need to use a ruler on the screen I think. The authors might consider if it is really necessary to include all 2x4 figures to show the same thing? If adding more variables to the equation did not materially alter HR for the predictors, then simply stating this in the text would be adequate for me at least. At least give it a thought. 

Comment 8:

Supplemental tables: Wouldn't it be nice for the reader to have exact information on what is included in the model instead of having to twist and turn? There is no reason to not be super informative and helpful, given that the manuscript is being considered for an online journal, and this is supplemental information. 

Comment 9: 

Result section: 

Quoting: "We next estimated the association between lifetime educational attainment and income at age 40 in Models 1A and 1B, respectively"

This is not correct, is it? The outcome is AMC?

Quoting: "Among females (Figure and Supplementary Table 1), having the lowest level of

education was initially associated with a nearly 5-fold increase in risk of AMC (HR=4.71; 95% CI

3.85, 5.77)."

Consider whether referring to the exact model is more precise than using "initially".

Comment 10:

The reporting and use of interactions on an additive and multiplicative scale: 

I am not a statistician and have to admit that I have had and still have problems wrapping my head around the use of reporting additive interactions as an alternative to multiplicative interaction: 

The researchers ask the question of whether the association between education and income with the risk of AMC differs according to familial risk of AUD. Such a question is most often answered in a multiplicative model with interaction terms, which is what the researchers write that they have done.

If the interaction term shows a difference, this answers the research question. It is important to show the results of the test, and stratified HRs is often included to help the reader get an overview of how the nature of the interaction plays out for each group in comparison to each other, and in comparison with the results of the main model. That is what I expected to see, but the authors chose to report interaction on the additive scale from the multiplicative model. That is probably ok, but I have some minor comments.

Is it correct that results in table S11 are the same as S10, only with the interaction term? Seems to be materially the same, with slight variation. 

The combination of what is presented in table S11 and S12 and the manuscript text, would you say that it provides the reader with a good understanding of how the association between income and AMC differs? The text explains it to me, and I can see from table S12 that the RERI and S looks different for education and income, but there is no guidance for me to understand the magnitude of the difference between the groups. That doesn't mean its wrong or anything, but perhaps help the reader a bit more in this regard. 

In addition, I agree with the authors that it is important not to make a big fuzz about small stuff. For example, if the group with familial AUD is for example very small, then it is not that important (from a public health perspective) that the association between low education or low income with the risk of AMC is more pronounced, in comparison with a scenario where this group is large. From a clinical point of view, and if it were treatment types or drugs involved, it would perhaps be clinical important for a practitioner to know about the interaction, even though the interaction involves only a very small group. As the authors includes this argument, I would really like to see how many that have a family risk of AUD calculated by the family matrix. Or maybe I missed it? 

Comment 11:

Quoting: "As noted above, previous research has found that drinking patterns vary by SEP, with

those of lower SEP more likely to engage in heavy consumption (11, 12, 13, 14, 16), which is

strongly linked to risk of AMC (22, 23, 24). Information on consumption levels is not available in

the Swedish nationwide registries, precluding our ability to test whether the associations

between education/income remain when accounting for heavy episodic drinking, for example.

However, our finding that these associations persist when controlling for AUD - itself closely

associated with consumption levels (25, 26) - is consistent with prior evidence that individuals

of lower SEP are at disproportionately high risk of AMC even after accounting for heavy drinking

(16)."

Heavy episodic drinking or binge drinking is the consumption of a lot of alcohol per occasion, normally defined as 5+ or 6+ units, corresponding to 60+ grams of pure alcohol per occation. To qualify as HED, the person has to drink about 2 grams per day on average, which corresponds to 5+ units once a month. Heavy episodic drinking exist as a drinking pattern from this lower bound and up until the average daily consumption moves into the area of chronic heavy drinking or heavy drinking, which may be defined as >60 grams per day, which in practice means that the person can have a heavy episode drinking every day. 

My point of writing this, is that the authors refers to heavy episodic drinking and heavy drinking/AUD without making a clear distinction about how they are defined. I also commented on this earlier with regards to the introduction. Chronic heavy drinking is probably much more strongly related to AMC, because the amount of alcohol is considerable and takes a big toll on the metabolizing organs of the body. You woudn't expect the same thing for a person binge drinking once a week? 

HED is very important in relation to alcohol-related accidents, and therefore also relevant for alcohol related mortality, but whether a given amount of alcohol consumed in the form of HED or spread over several days conveys a different effect on organs, and AMC, is not a research question that has not been studied extensively with good methods. 

Perhaps better to only focus on AUD/heavy drinking/alcohol intake measured as the quantity, rather than mixing in HED in the discussion. Consider. I think maybe that this discussion regarding alcohol intake should be made after the discussion that adjustment for AUD registrations did only slightly moderate the association of SEP with AMC. Could it be that the measurement of AUD is not very good a picking up alcohol intake that can lead to AMC? That is my interpretation. Any other interpretation would imply a link between SEP and AMC via another route than alcohol?

Comment 12:

Quoting: "Third, we found that individuals with higher genetic liability to AUD were more

susceptible to the adverse impact of low income, though this was not the case for those with

lower levels of educational attainment. This could reflect the stronger overall effect size of

income level, and replication in other samples, and perhaps using complementary methods

(e.g., molecular polygenic scores) is needed."

Consider including the information that the test for interaction was performed at year 0, when the magnitude of the association of income with AMC was most pronounced. 

Comment 13:

Quoting: "Finally, alcohol consumption varies across countries and cultures; the current findings might not generalize to other contexts."

I agree that this study is irrelevant for cultures where alcohol is not consumed, but this is obvious and does not leave the reader from another country with any meaningful insight about whether this study applies to their country. And alcohol consumption is not the only thing that can influence the external validity of these findings. A thorough discussion regarding the external validity of the findings is lacking in the current manuscript. 

Take women for example, as the results are stratified by gender. They are likely to vary more between countries then men in terms of their alcohol intake, their educational attainment at age 40, and whether they work or not at age 40, for example depending on the access to kindergarten at a subsidized price. In Canada for example, the price for child care at day time is not the same as in Norway, which could contribute to more women choosing to stay home for more years. These are only thoughts.

Reviewer #3: This is a useful study based on a very large national cohort. The conclusions are well-supported by the data, and the strengths are many.

The authors refer to the outcome variable as "alcohol-related medical conditions". This is technically correct, but to add precision, I would recommend using the term "fully alcohol-attributable disorder or disease". As far as I could see from the list of diagnoses in the text, all values would qualify to be fully AADDs. That way, nobody would think that the article is about AUDs or broader health conditions that could be influenced by alcohol, such as cancer or stroke.

In the abstract, the authors present the findings that are not based on genetic risk first, then the ones that are. However, the interaction with genetic risk is more cutting edge and has less uncertainty, so I would recommend that these are given focus in both the abstract and the title.

The analysis treats death and emigration as similar to regular censoring. This will lead to biased estimates. The standard today is to use competing risks models, the type that was first developed by Fine and Gray (see Austin et al 2022 for an updated discussion). A competing risks regression model may not be estimable with such a large dataset, but perhaps the authors can run an analysis on a random sample of the population to estimate the influence of death. However, without considering death as a competing risk, the decline in risk associated with low education could be a complete artefact: it just seems that the risk declines over time, because those with low income and low education die disproportionately at a younger age, and are no longer at risk of AMD. I may be completely wrong here, but if so, I think the authors should explain why.

The authors write that "The polychoric correlation between income quartile and education level for females was 0.185 (SE-0.001) and for males was 0.198 (SE=0.001), suggesting that while these socioeconomic measures are related, they also provide non-overlapping information." This is not a very formal test of the degree of dependence or independence of the two variables. I think I get that the authors want to argue that both variables should be considered (and I agree), while they also want to acknowledge that both come under a common heading, SES (again - I agree). But if they want to provide a statistical argument for degree of dependence, I would want something stronger.

Reference

Austin PC, Putter H, Lee DS, Steyerberg EW. Estimation of the Absolute Risk of Cardiovascular Disease and Other Events: Issues With the Use of Multiple Fine-Gray Subdistribution Hazard Models. Circ Cardiovasc Qual Outcomes. 2022.

[LINK]

---

## [Decision Letter · Decision Letter 2]

9 Feb 2024

Dear Dr. Edwards,

Thank you very much for re-submitting your manuscript "Socioeconomic position indicators and risk of alcohol-related medical conditions: An observational Swedish national cohort study" (PMEDICINE-D-23-03228R2) for review by PLOS Medicine.

I have discussed the paper with my colleagues and the academic editor and it was also seen again by 2 reviewers including the statistical reviewer. I am pleased to say that provided the remaining editorial and production issues are dealt with we are planning to accept the paper for publication in the journal.

[LINK]

We look forward to receiving the revised manuscript by Feb 16 2024 11:59PM.   

Sincerely,

Philippa Dodd, MBBS MRCP PhD

Senior Editor 

PLOS Medicine

plosmedicine.org

COMMENTS FROM THE ACADEMIC EDITOR:

I agree with the decision to proceed.

I had just one more suggestion: Might it be helpful to clarify in the abstract that these results are from a high-income country? I would hypothesize that these risks would be very different in low-income countries. Sweden is a unique country in many ways.

COMMENTS FROM THE EDITORS:

GENERAL

Thank you for your detailed and considered responses to previous editor and reviewer comments. Please see below for further comments which we require you address prior to publication.

TITLE

Thank you for revising your title, we suggest, ‘Socioeconomic position indicators and risk of alcohol-related medical conditions: A national cohort study from Sweden.’

ABSTRACT

Abstract methods and findings - ‘N=4253 (0.37%) of females and N=11,183 (0.93%) of males…’ please remove the ‘N=’ and the ‘of’ on both occasions here to improve reader accessibility.

AUTHOR SUMMARY

Line 3 – suggest ‘…excess morbidity and mortality…’ instead.

Please add a final bullet point after line 24 of ‘What Do These Findings Mean?’, which briefly describes the main limitations of the study in non-technical language.

INTRODUCTION

Page 6, line 5 – should the colon be a full stop?

METHODS and RESULTS

Results page 12 onwards - Multivariable models – if possible, we think that the presentation/description of your results could be made more concise to improve reader accessibility. It is not difficult to get a bit lost in the detail. Please revise for the purpose of improved brevity.

SOCIAL MEDIA

To help us extend the reach of your research, if not already done so, please detail any X (formerly Twitter) handles you wish to be included when we tweet this paper (including your own, your coauthors’, your institution, funder, or lab) in the manuscript submission form when you re-submit the manuscript.

COMMENTS FROM THE REVIEWERS:

Reviewer #1: The authors have addressed my points.

Michael Dewey

Reviewer #2: To the authors, 

I did not identify or raise any major concerns about the integrity of the study when reading the manuscript for the first time but had a list of questions and comments for the authors to clarify and consider, respectively. Most of which were of minor importance. The authors have addressed them thoroughly in their rebuttal and made changes to the manuscript text and supplemental methods and tables as a result, which I find improves clarity. I have not identified any new concerns when reading the revised manuscript and have no further comments or suggestions on how to improve the manuscript. 

Eirik Degerud

[LINK]

---

## [Editor Report · Decision Letter 3]

14 Feb 2024

Dear Dr Edwards, 

On behalf of my colleagues and the Academic Editor, Dr David Flood, I am pleased to inform you that we have agreed to publish your manuscript "Socioeconomic position indicators and risk of alcohol-related medical conditions: A national cohort study from Sweden" (PMEDICINE-D-23-03228R3) in PLOS Medicine.

PRESS

Thank you again for submitting to PLOS Medicine, it has been a pleasure handling your manuscript. We look forward to publishing your paper. 

Kind regards,

Pippa

Philippa C. Dodd, MBBS MRCP PhD 

PLOS Medicine

pdodd@plos.org